# Lactoferrin as a Candidate Multifunctional Therapeutic in Synucleinopathies

**DOI:** 10.3390/brainsci15040380

**Published:** 2025-04-06

**Authors:** Caroline A. Barros, Tuane C. R. G. Vieira

**Affiliations:** Institute of Medical Biochemistry Leopoldo de Meis, National Institute of Science and Technology for Structural Biology and Bioimaging, Federal University of Rio de Janeiro, Rio de Janeiro 21941-599, RJ, Brazil; carol.augustobarros@gmail.com

**Keywords:** lactoferrin, alpha-synuclein, aggregation, synucleophaties, Parkinson’s disease

## Abstract

Lactoferrin (Lf) is a multifunctional glycoprotein with well-established antimicrobial, anti-inflammatory, and iron-binding properties. Emerging evidence suggests that Lf also plays a neuroprotective role, particularly in neurodegenerative disorders characterized by protein aggregation, such as Parkinson’s disease (PD). Alpha-synuclein (aSyn) aggregation is a pathological hallmark of PD and other synucleinopathies, contributing to neuronal dysfunction and disease progression. Recent studies indicate that Lf may interfere with aSyn aggregation, iron chelation, and modulation of oxidative stress and neuroinflammation. Additionally, Lf’s ability to cross the blood-brain barrier and its potential impact on the gut-brain axis highlight its promise as a therapeutic agent. This review explores Lf’s mechanisms of action in synucleinopathies, its potential as a disease-modifying therapy, and innovative delivery strategies that could enhance its clinical applicability. By addressing the pathological and therapeutic dimensions of aSyn aggregation, we propose Lf as a compelling candidate for future research and clinical development in neurodegenerative diseases.

## 1. Introduction

Lactoferrin (Lf) was first identified in 1939 by Sorensen and Sorensen as the red protein in whey. Later, in 1960, it was isolated and purified from human and bovine milk [1]. Lf is a glycoprotein found in milk and other bodily fluids, including saliva, semen, and blood. It is multifunctional and plays a role in various biological processes. Lf is widely recognized for its numerous functions, which include anti-inflammatory, antifungal, antitumor, immunomodulatory, and antimicrobial activities [2]. This characteristic is also observed in several mammals, with high sequence identity and significant conservation of its functional structure [3].

Lf plays a significant role in both neonates and older adults. It is responsible for anti-infective, immunological, neurodevelopment, and gastrointestinal activities in young children. Additionally, Lf is important for iron absorption, cell proliferation, intestinal cell maturation, and enhancing the intestinal barrier [4,5,6]. It constitutes about 20% of the proteins found in human milk and is produced in high concentrations in milk and colostrum, ranging from 1 g/L to 7 g/L, respectively [7].

During aging, Lf offers essential benefits, including anti-inflammatory effects and the ability to neutralize reactive oxygen species (ROS), which helps protect cells [7]. Studies have demonstrated that Lf supplementation can enhance the innate immune system in elderly individuals [8].

In recent years, various studies have explored the potential use of Lf as a nanoparticle carrier, especially in treating cancer [9]. Consequently, additional applications have emerged, including the possible use of Lf and its properties in addressing neurodegenerative diseases.

Lf is recognized for its essential roles in immune function and antimicrobial defense. It has also shown potential in addressing protein misfolding and aggregation, which are significant contributors to amyloidosis. This review examines the role of Lf in neuroprotection, with a focus on synucleinopathies such as Parkinson’s disease (PD). The findings suggest that Lf may be a promising option in the search for effective therapies.

## 2. Lactoferrin as a Novel Frontier in Synucleinopathy Therapies

### 2.1. Evolutionary Versatility

Lf is milk’s second most abundant protein, following caseins. Its concentration varies depending on the species and stage of lactation [2,10]. Beyond milk, Lf is found in various biological fluids, including tears, saliva, blood, semen, vaginal secretions, nasal and bronchial fluids, bile, gastrointestinal secretions, urine, and amniotic fluid. It is also a key component of the secondary granules of neutrophils [10].

Excess-free iron in the bloodstream and bodily fluids can lead to the formation of ROS, causing oxidative stress and cellular damage. Transferrin family proteins primarily regulate iron levels. These proteins are categorized based on their presence in different biological environments: serum transferrins (Tfs) in the bloodstream, Lf in secretions, and ovotransferrin (oTf) in egg whites [11,12,13,14,15].

Various mammals produce Lf, including humans, horses, cattle, goats, dogs, and rodents. Across mammalian species, Lf shares approximately 70% sequence identity. For example, human and mouse Lf exhibit 70% homology, human and bovine Lf share 69%, mouse and bovine Lf have 63% similarity, and human and chimpanzee Lf display the highest homology at 97% [2,10,16].

Lf has evolved to serve many functions beyond iron sequestration, including antibacterial, anti-inflammatory, antitumor, immunomodulatory, antiviral, and antifungal properties, among others [17,18].

As a potent immunomodulator, Lf influences inflammatory processes and adaptive and innate immune responses. It is stored in the secondary granules of neutrophils and exhibits a high affinity for receptors found on macrophages, monocytes, and lymphocytes. Lf modulates the migration and activation of antigen-presenting cells and the expression of cytokines and chemokines, highlighting its role in orchestrating immune responses [19,20,21].

The antitumor effects of Lf have been documented against various cancer cell types, including glioma, human colon adenocarcinoma, and prostate cancer [9,22,23,24]. Notably, in the case of human colon adenocarcinoma, Lf exerted its antitumor effects by inhibiting angiogenesis, thereby suppressing tumor growth [22]. Lf has also been explored as a potential nanocarrier in liposomal formulations to enhance its therapeutic efficacy. In a study by Serrano et al. (2020), 75 patients treated with a liposomal containing Lf (32 mg), zinc (10 mg), and vitamin C (12 mg) experienced symptom improvement within 4 to 5 days of treatment [25].

Lf’s antiviral capabilities have attracted significant attention, particularly in the context of the COVID-19 pandemic. In vitro studies have demonstrated that Lf can inhibit the infection of both SARS and SARS-CoV-2 in a dose-dependent manner. This effect is attributed to Lf’s interaction with heparan sulfate proteoglycans on the cell surface, which blocks viral entry [26,27]. Additionally, in silico experiments have suggested that Lf can bind to the SARS-CoV-2 spike protein, further preventing the virus from entering host cells [26].

Beyond SARS-CoV-2, Lf is known for its broad-spectrum antiviral activity against a range of viruses, including Zika, chikungunya, dengue, Mayaro, herpes simplex, rhinovirus, and influenza A [28,29,30,31,32,33]. Despite the challenges associated with Lf’s bioavailability, especially with oral administration, its incorporation into nanocarriers or combination with other bioactive compounds appears promising for enhancing its clinical efficacy.

### 2.2. Lactoferrin Structure and Iron Binding

Lf is an iron-binding protein that belongs to the transferrin family. It exists in several forms: the apo form, which contains little to no iron (0–6% saturated); the native form, which is partially saturated with iron (10–20%); and the fully iron-saturated form, known as holo, which is between 76% and 100% saturated [2].

The structure of Lf consists of two symmetrical lobes, the N-lobe and the C-lobe. A flexible hinge region connects these lobes of 11 amino acids arranged in an alpha-helix, which is susceptible to proteolytic cleavage. This hinge region is crucial to Lf’s ability to bind and release iron ions. Each lobe is divided into two similar domains: the N-lobe contains the N1 and N2 domains, while the C-lobe includes the C1 and C2 domains [10] (Figure 1).

The iron-binding site in Lf consists of four amino acid residues: one aspartate, two tyrosines, and one histidine. It also involves synergistic interactions with a carbonate ion (CO_3_^2−^), two phenolates, and one imidazole nitrogen. The affinity constant of Lf for Fe^3+^ is approximately 10^20^, while for Fe^2+^ it is around 10^3^ [16,34] (Figure 1). These anions involved in Fe^3+^ binding are essential because they neutralize the positive and negative charges in the surrounding environment. Lf can also bind to other metal ions such as Cu^2+^, Zn^2+^, Mn^2+^, Al^3+^, Ga^3+^, and Co^3+^, but the association of Lf with these ions occurs under specific conditions and with lower affinity [10,16,35].

The interaction of iron ions in the Lf structure causes a change in its conformation [36]. In the holo-lactoferrin (holo-Lf) form, the lobes are more closed, while the apo-lactoferrin (apo-Lf) structure is more dynamic, flexible, and open. Interestingly, even without iron saturation, apo-Lf can still exhibit closed lobes. This phenomenon occurs because the two states have a small energetic difference, allowing for structural fluctuation [16].

The N-terminal region of Lf exhibits a positive charge distribution, particularly in the interlobular region and on the outer part of the first α-helix. This high positive charge results in an isoelectric point (pI) of approximately 9, enabling Lf to interact with various anionic partners, such as DNA and glycosaminoglycans [2,16].

Lf’s structural characteristics enable it to interact with various proteins and receptors, contributing to its diverse effects. It binds to Lf receptors, regulating cellular iron uptake and distribution, which prevents iron overload and reduces oxidative stress caused by free iron [37,38]. It also binds to the Low-Density Lipoprotein Receptor-Related Protein 1 (LRP1), which plays a critical role in clearing amyloid-beta (Aβ) peptides that accumulate in Alzheimer’s disease (AD) [39,40,41]. One intriguing hypothesis is that these interactions may interfere with toxic aggregation pathways, positioning Lf in neuroprotection.

## 3. Alpha-Synuclein Aggregation: A Target Ripe for Disruption

Synucleinopathies represent a group of neurodegenerative disorders characterized by the abnormal accumulation of alpha-synuclein (aSyn) in the brain. These diseases, including PD, dementia with Lewy bodies (DLB), and multiple system atrophy (MSA), are associated with progressive neurological decline, highlighting their clinical and societal significance [42].

PD is primarily characterized by motor symptoms resulting from the loss of dopaminergic neurons [43]. DLB shares these symptoms but also presents early cognitive decline and hallucinations [43]. In contrast, MSA involves more widespread neurodegeneration, with severe autonomic dysfunction and faster progression [43,44].

aSyn, a small, highly conserved protein, is predominantly found in presynaptic terminals, contributing to synaptic function and neurotransmitter release [45]. However, its propensity to aggregate under pathological conditions has been implicated as a major driver of neurodegeneration. These aggregates disrupt cellular processes and ultimately lead to neuronal death [45,46], underscoring the importance of addressing aSyn aggregation to mitigate synucleinopathies.

aSyn is an intrinsically disordered protein lacking a stable secondary or tertiary structure under physiological conditions. While critical for its normal function, this structural flexibility makes it prone to misfolding. Environmental factors, such as oxidative stress, post-translational modifications, and interactions with lipid membranes, can destabilize aSyn and promote its transition into pathogenic conformations [47,48,49,50].

The initial misfolding of alpha-synuclein sets off a cascade of aggregation (Figure 2). Misfolded monomers interact to form oligomers—small, soluble aggregates that are particularly toxic to neurons. These oligomers disrupt cellular membranes, impair calcium homeostasis, and initiate inflammatory responses [46,48,51]. Over time, oligomers coalesce into insoluble fibrils, which accumulate as Lewy bodies, a pathological hallmark of synucleinopathies [52].

Efforts to prevent aSyn aggregation include the development of small molecules, peptides, and antibodies. Small molecules, such as molecular chaperones, stabilize the native structure of aSyn. Peptides, monoclonal antibodies, and activators of autophagy and proteasome pathways target aggregated forms, neutralizing their toxic effects [53].

Neuroprotective agents, such as antioxidants and mitochondrial stabilizers, are being explored to mitigate the downstream effects of aSyn aggregation. These approaches enhance neuronal resilience and prevent cell death, offering potential symptomatic and disease-modifying benefits [53].

### Iron and Alpha-Synuclein Aggregation

Iron dysregulation is a factor in the development and progression of PD [54,55] (Figure 3). Iron imbalance leads to neurodegeneration through oxidative stress and ferroptosis, an iron-dependent form of programmed cell death [55,56]. This intricate relationship exacerbates the symptoms and progression of PD through interconnected biological processes.

Iron’s role as a redox agent makes it a key contributor to producing ROS, which causes oxidative stress and neuronal damage [55,57]. Increased iron levels in the brains of PD patients are documented [58,59,60,61,62,63], with aberrant iron deposition linked to oxidative stress and the promotion of aSyn increase [64] and aggregation [50,65,66,67]. Oxidative stress damages cellular components and creates a toxic environment conducive to the misfolding and aggregation of aSyn [68,69], amplifying neurodegeneration. In turn, increased aSyn expression enhances dopaminergic neurons’ sensitivity to ferroptosis [70]. aSyn aggregates disrupt iron metabolism, increasing iron accumulation and neuron redistribution, thereby promoting ferroptosis [71,72,73,74]. The phosphorylation of aSyn linked to disease (in Y125 and S129) enhanced its affinity to ferrous iron species [75]. Additionally, research indicates that iron is involved with aSyn spreading pathology [66,76], where misfolded aSyn propagates between cells, suggesting a systemic effect that may require broader metabolic interventions.

While iron dysregulation is undoubtedly a key, this growing understanding of iron’s role in aSyn aggregation has opened new avenues for therapeutic strategies to restore iron homeostasis. Iron chelators, for instance, show promise in slowing PD progression by reducing iron-induced toxicity and oxidative stress. Iron chelators decreased the levels of aSyn [55] and demonstrated protective effects in PD models [65,77]. Deferiprone (DFP), an iron chelator, completed phase 2 trials for PD treatment (NCT02655315, NCT01539837, NCT02728843) [78,79,80]. Randomized controlled trials demonstrated that DFP significantly decreased iron buildup in the brains of PD patients, although it did not change their clinical presentation [80]. Its lack of impact on clinical symptoms highlights the complexity of PD pathogenesis. This underscores the need for further investigation into combination therapies or additional approaches that not only target iron dysregulation but also address other contributing factors to neurodegeneration and symptom progression in PD.

By deepening our understanding of the relationship between iron metabolism, aSyn aggregation, and neurodegeneration, we can get closer to developing effective therapies that slow disease progression and enhance the quality of life for individuals with PD. Can a naturally occurring protein counteract the pathological aggregation of aSyn? Given its diverse functional capabilities, Lf stands out as a potential candidate to target specific stages of the aSyn aggregation process, toxic effects, and iron metabolism, providing a new perspective on addressing this complex pathology.

## 4. Lactoferrin in Parkinson’s Disease

Lf is related to aging and, consequently, to its diseases, such as AD and PD. In a healthy individual, the concentration of Lf in the cerebral cortex is virtually null. However, this concentration increases during aging and the onset of neurodegenerative diseases, such as PD (Table 1) [81,82,83].

Some studies have shown the influence of Lf in PD using mouse models of PD treated with MPTP (1-methyl-4-phenyl-1,2,3,6-tetrahydropyridine) [83,84,86]. In these mice, the MPTP toxin is converted into its activated form, MPP+ (1-methyl-4-phenylpyridine), which is known to inhibit proper mitochondrial function, impairing cellular respiration. Treatment with MPTP leads to the loss of dopaminergic cells, resulting in motor and behavioral deficits in the animals, similar to what occurs in PD [89].

The study by Wang and colleagues (2015) demonstrated that under conditions of increased iron load, the expression of Lf is positively regulated by activated microglia. Lf was able to mitigate MPP+-mediated neurotoxicity in neurons, and this improvement was not related to its iron saturation. Instead, it was linked to its influence on enhancing Cu/Zn-superoxide dismutase activity, thereby exerting a beneficial effect on oxidative stress and increasing the expression of BCL-2, an anti-apoptotic protein [83].

A significant reduction in the toxicity mediated by this toxin was observed in mice that received Lf as a pre-treatment before MPTP injection. The pre-treatment was associated with several benefits, such as less body weight loss, partial recovery of tyrosine hydroxylase-positive cells (an essential enzyme for dopamine production) in the substantia nigra and striatum, as well as improvement in motor functions and signs of neuroregeneration [86]. Furthermore, another study demonstrated that Lf suppressed the excessive iron accumulation induced by MPTP and the upregulation of divalent metal transporter (DMT1) and transferrin receptor (TFR), key proteins in intracellular iron regulation. As a result, there was an improvement in the activity of several antioxidant enzymes [85].

A recent study demonstrated that the deficiency of Lf in mice astrocytes, achieved by specifically deleting the Lf gene in these cells, reduced the number of tyrosine hydroxylase-positive cells. This also caused mitochondrial dysfunction in the remaining dopaminergic neurons, resulting in motor deficits. Notably, these deficits were further aggravated after treatment with MPTP [87]. In both in vitro and in vivo models, abnormal effects related to glutamate and calcium homeostasis, mitochondrial structure and function, mitochondrial dynamics, and the relationship between mitochondria and endoplasmic reticulum membranes were observed. These changes were accompanied by indications of oxidative stress and endoplasmic reticulum stress, which contribute to the increased vulnerability of dopaminergic neurons [87]. This study indicates that Lf affects dopaminergic neurons by regulating glutamate and calcium metabolism and endoplasmic reticulum-mitochondria signaling, suggesting it could be a promising target for PD treatment [87].

Research involving MPTP-treated mice, which serve as a model for PD, has demonstrated that Lf treatment offers significant protection against neurodegeneration. The studies indicate that Lf treatment reduces iron dysregulation, improves motor function, and addresses behavioral disorders. Additionally, Lf acts as an antioxidant, stimulates the expression of antioxidant proteins, and provides anti-apoptotic effects on cells [37,85].

Chronic exposure to rotenone, a pesticide, plays a significant role in the development of PD by damaging dopaminergic neurons in the nigrostriatal system [88,90]. Mice treated with rotenone display characteristics similar to those of PD, including the formation of Lewy bodies [91]. Furthermore, rotenone can lead to chronic toxicity and increase the risk of developing PD in humans. Previous studies have demonstrated that in a rotenone-induced cellular model of PD using SH-SY5Y cells, pre-treatment with Lf significantly reduced the loss of cell viability, diminished impairment of mitochondrial membrane potential (MMP), and lowered ROS generation [88]. Additionally, in a mouse model treated with rotenone, Lf mitigated neurodegeneration in the substantia nigra and improved motor deficits observed in the rats [88,92].

The study by Hamm-Alvarez et al. (2019) investigated the use of reflex tears as a diagnostic tool for PD [93]. The results showed a significant increase in LF levels in the tears of PD patients compared to the control group (196.05 µg/mg vs. 145.84 µg/mg, *p* value 0.002). This finding suggests that LF could be a promising biomarker for PD [93].

Lf has been associated with amyloid deposits in the brain, but it is still unclear whether it can directly bind to amyloid structures to prevent or reduce their aggregation. There is a possibility that Lf acts in the early stages of aSyn aggregation, directly interacting with the aSyn monomer to prevent its aggregation, as well as interacting with other oligomeric species, such as oligomers and protofibrils, functioning as a disaggregase (Figure 2).

Studies suggest that through electrostatic interactions, Lf may inhibit aggregation by forming soluble complexes with denatured proteins, such as alpha-lactalbumin, and heparin and nucleic acids. This effect may be related to the charge distribution in Lf’s structure [94].

Although full-length Lf cannot aggregate under physiological conditions, a peptide derived from Lf, with the sequence NAGDVAFV, has been identified as highly amyloidogenic [95]. Moreover, Lf seems to associate with the surface of the peptide fibril, forming mixed peptide/protein fibrils. However, there is no evidence that Lf alone can form exclusively Lf fibrils. Additionally, it has been suggested that the interaction between Lf and the peptide is selective and that the process does not involve a seeding mechanism [95].

Parkin protein is a member of the RBR (RING-between-RING) E3 ubiquitin ligases and degrades disordered and redundant proteins [96,97]. Mutations in the parkin gene are the most common causes of familial PD and autosomal recessive juvenile PD. Studies have reported that aggregated aSyn can promote the abnormal accumulation of parkin and, to a lesser extent, α-tubulin, leading to alterations in the neuronal cytoskeleton [98]. Furthermore, it has been shown that parkin also binds to Lf and ubiquitylates it, positively influencing iron homeostasis [99]. These findings suggest that parkin is crucial in regulating cellular iron and maintaining neuronal cytoskeleton integrity. Given that both parkin and Lf contribute to iron homeostasis, and aSyn aggregation disrupts these processes, a complex interplay between these proteins may exist in PD. Thus, the parkin-aSyn-Lf axis may represent a critical regulatory network in PD, where disruptions contribute to iron imbalance and proteotoxic stress, highlighting potential therapeutic targets to restore neuronal homeostasis.

## 5. Lf in Other Neurodegenerative Disorders

### 5.1. Lf in Alzheimer’s Disease

In AD, Lf can be found in the brain’s limbic system, associated with senile plaques and neurofibrillary tangles [82,100]. The study by Wang and colleagues (2010) showed through immunostaining that Lf is highly expressed and accumulates in senile plaques in older mice (>18 months) [100].

It has already been demonstrated that Lf treatment for three months in patients with AD showed an improvement in serum levels of anti-inflammatory markers, with a decrease in the pro-inflammatory cytokine IL-6, an increase in the anti-inflammatory cytokine IL-10, and an increase in antioxidant markers. It was also shown that there was a decrease in serum levels of oxidative stress markers, cholesterol, caspase-3, and phosphorylated tau protein, among others [101]. AD patients also showed improvement in cognitive decline when treated with Lf. Although it is a pilot study, this work suggests that the effect of Lf in AD could provide essential insights into how Lf might help in this pathology [101].

A recent study investigated the effects of astrocytic Lf overexpression in the brains of Alzheimer’s model mice (APP/PS1) [102]. The overexpression of Lf in astrocytes improved cognitive capacity and reduced neuronal loss by decreasing iron accumulation and increasing glutathione peroxidase 4 (GPX4) expression in neurons. Treatment with human Lf (hLf) also inhibited ferroptosis induced by ammonium ferric citrate (FAC) by chelating intracellular iron. Additionally, it was observed that hLf increased the internalization of LRP1 and facilitated its interaction with heat shock cognate protein 70 (HSC70), which inhibited HSC70 binding to GPX4, preventing its degradation and attenuating FAC-induced ferroptosis [102].

The processing of amyloid precursor protein (APP) generates the amyloid-β (Aβ) peptide, which forms the extracellular plaques characteristic of AD [103]. Despite the described benefits of Lf in neurodegenerative diseases, one study reported that holo-Lf interacts with APP, secreted by activated microglia, which may affect amyloidogenic processing and contribute to Aβ plaque formation. Pharmacological inhibition of this interaction, while maintaining the neuroprotective effects of Lf, could be a potential therapeutic strategy for AD, but further studies are needed to clarify the role of this interaction [103].

Recent studies suggest that salivary lactoferrin may be a promising biomarker for AD and amnestic mild cognitive impairment [104,105]. Carro et al. (2017) observed a correlation between decreased levels of Lf in saliva and the presence of AD and MCI, indicating that saliva, rich in proteins derived from the blood and central nervous system, may reflect significant biological changes [104]. Furthermore, 14 of the 18 individuals in the control group developed mild cognitive impairment or AD, reinforcing the hypothesis that Lf could be used to identify early stages of the disease [104]. On the other hand, a study by Gleerup et al. (2021) found no association between Lf levels in saliva and cerebrospinal fluid (CSF) in patients with AD and other neurological conditions, such as vascular dementia and PD [106]. The difference in results may be attributed to the homogeneity of Carro’s group compared to the more significant heterogeneity of Gleerup’s group [107]. Supporting the idea that salivary Lf has diagnostic potential, González-Sánchez et al. (2020) showed that Lf could distinguish between AD, prodromal AD, and frontotemporal dementia with high sensitivity and specificity [105]. Additionally, research in mice also identified a reduction in salivary Lf, suggesting that dysregulation of salivary glands may contribute to this decrease in AD. These findings indicate that more studies are needed to validate Lf as a non-invasive, low-cost, and reliable biomarker for AD diagnosis [108].

### 5.2. Lf in Prion Disease

Given the various therapeutic activities described for Lf, some researchers have been interested in evaluating its potential antiprion activity. The first study aimed to understand whether Lf could have antiprion activity by assessing its effects on scrapie prion protein (PrP^Sc^) accumulation in ScN2a58 cells [109]. The results showed that Lf inhibits the accumulation of PrP^Sc^ in a dose- and time-dependent manner, reducing the infectivity of the prion protein in ScN2a58 cells and suggesting that Lf can interact with both cellular prion protein (PrP^C^) and PrP^Sc^ through immunoprecipitation experiments. This study indicates that understanding how Lf acts on prion proteins may contribute to developing a therapeutic agent [109]. However, the interaction between Lf and prion proteins and the cellular localization of PrP in response to Lf treatment requires further investigation. To address this, various techniques, such as spectroscopic methods and confocal fluorescence microscopy, could be used to determine cellular localization and provide a more detailed understanding of how this interaction occurs [109].

Another study used a PrP (106-126) peptide to model mitochondrial dysfunction in neuronal cells [110]. Mitochondrial dysfunction can lead to apoptosis through the accumulation of ROS, which in turn leads to neurodegeneration. Since Lf is known for its antioxidant properties, its role in mitochondrial dysfunction caused by PrP (106-126) was investigated [110]. The results showed that Lf protects against ROS production in human neuroblastoma cells (SH-SY5Y) and that treatment with Lf inhibits JNK and caspase-3, preventing the cells from undergoing apoptosis [110]. Although this study demonstrated the role of Lf in mitochondrial dysfunction, the data were indirect, as the influence of PrP (106-126) on pathways affected by oxidative stress was examined indirectly. More direct approaches to assess mitochondrial dysfunction, such as measuring oxygen flow variations, citrate synthase activity, and evaluating the damaging effects of oxidative stress on DNA and lipids, would provide more definitive evidence [110].

Subsequently, another study aimed to identify the effect of Lf on the enzymatic activity and expression of PHD2, inhibiting hypoxia-inducible factor alpha-1 (HIF-1α), leading to cell death [111]. HIF-1α regulates the expression of PrP^C^, and the addition of PrP (106-126) induces cell death. This study showed that Lf could regulate the activity of PHD2, thus inhibiting the degradation of HIF-1α and preventing cell death in the presence of PrP (106-126). The authors suggest that investigating Lf as a potential therapy for inhibiting PHD2 could be valuable [111].

## 6. Beyond the Brain: The Gut-Brain Axis

The gut microbiota is crucial in regulating the central and autonomic nervous systems [112]. Gastrointestinal dysfunction, as intestinal dysbiosis, is a common symptom in PD and can appear years before the onset of the characteristic motor symptoms of the disease [112,113,114]. Furthermore, intestinal dysbiosis or its products are directly involved in the aggregation process of aSyn. The gut-brain microbiota axis has become increasingly relevant in PD research, as growing evidence suggests a link between microbiota alteration and disease progression.

The intestinal epithelium is a protective barrier against pathogen invasion [115]. However, when this barrier is compromised, it can trigger a series of positive feedback loops [116]. This intestinal dysfunction significantly alters the microbiota, promoting the growth of pro-inflammatory species [117]. As a result, intestinal inflammation is exacerbated, increasing reactive oxygen and nitrogen species within the intestinal lumen [118]. This leads to greater mucosal permeability, oxidative stress, and inflammatory responses. These conditions are directly linked to the aggregation of aSyn in the enteric nervous system (ENS) [119].

Studies have shown that patients with PD exhibit a reduction in the abundance of several beneficial bacteria, such as the families *Prevotellaceae* and *Lachnospiraceae*, and the genera *Prevotella* (particularly *Prevotella copri*), *Ruminococcus*, *Blautia*, *Dorea*, *Roseburia*, and *Faecalibacterium* [119]. In contrast, there is an increase in pathogenic bacteria, particularly gram-negative species like the families *Enterobacteriaceae* and *Verrucomicrobiaceae* and the species *Escherichia coli* [119]. These alterations in bacterial composition reflect the intestinal dysbiosis associated with PD and may contribute to disease progression [119].

Lf is a protein that plays a significant role in defending against pathogenic bacteria and also helps promote certain beneficial prebiotic bacteria in the gut. Research has shown that dietary Lf can positively impact gut microbiota balance in mouse models of AD [120]. Additionally, Lf exerts a protective effect on the intestinal barrier, with reports showing that it improves the barrier function of a Caco-2 cell layer damaged by lipopolysaccharide (LPS) [121].

In a mouse model of AD with ulcerative colitis, Lf was able to alter the gut microbiota significantly, improving the cognitive performance of the mice [122]. A case-control study evaluated whether fecal markers of intestinal inflammation and permeability could be used as predictors for identifying PD patients at higher risk of developing intestinal inflammation. Lf, beta-defensin, and zonulin were at higher levels than in the control group. Additionally, Lf and beta-defensin showed a potential correlation in predicting the likelihood of PD compared to controls [123].

In this context, Lf emerges as a promising protein, not only for its role in defending against pathogenic bacteria but also for its ability to promote beneficial prebiotic bacteria and protect the intestinal barrier, which may have therapeutic implications in modulating the microbiota and protecting the gastrointestinal tract in patients with PD.

## 7. Innovative Applications: Delivery Systems and Combinatorial Strategies

### 7.1. Target Delivery

The blood-brain barrier (BBB) is a natural, semi-permeable protective structure that surrounds the microvasculature of the central nervous system [124]. It is composed of tightly connected endothelial cells lining the inner walls of the vessels [124,125]. This barrier regulates the entry and exit of ions, molecules, and cells between the vascular compartment and the brain [125].

Effective delivery across the BBB continues to pose a significant challenge in treating brain tumors, cerebrovascular diseases, and neurodegenerative disorders [125]. Recent advancements in drug delivery systems, such as nanoparticle encapsulation and intranasal administration, offer promising solutions to enhance the bioavailability of Lf in the brain [126,127,128]. Nanoparticles, which typically range in size from 10 to 1000 nm in diameter, present several advantages for drug delivery, including reduced drug dosage, low toxicity, and safe, targeted delivery to the intended organ [129].

Currently, several studies have demonstrated the possibility of using Lf as a specific targeting vector, with the brain as the target, due to its high affinity for LfRs located in the endothelial cells of the cerebral capillaries and the BBB [130,131,132,133,134,135,136,137,138,139]. A study investigated the mechanism of Lf transport to the brain and found that differentiated bovine brain capillary endothelial cells had both high- and low-affinity binding sites for Lf [81]. Only the differentiated cells were able to internalize Lf, suggesting that Lf receptors are acquired during cell differentiation [81]. The transport of Lf occurred in a unidirectional manner, receptor-mediated, with no apparent intracellular degradation. Additionally, it was observed that iron can cross endothelial cells in complex with Lf [81].

The receptors are found not only in the cells that make up the BBB but are also expressed in glioblastoma cells [140]. Consequently, some studies explore whether Lf-associated nanoparticles can cross the BBB and have an antitumor effect [140,141]. Additionally, another approach under investigation involves using Lf-loaded nanoparticles for treating neurodegenerative diseases and as antiretrovirals [126,127,128,130,132,140,141,142,143].

Lf has been used to deliver various drugs, including puerarin, deferasirox, curcumin (which helps reduce the formation of Aβ plaques), nerve growth factor, coumarin 6, S14G humanin (a neuroprotective agent), and huperzine A. These drugs are aimed at treating diseases such as AD and PD. Increased expression of LfRs has been observed, highlighting the significance for Lf-associated drugs in treatment [27,144,145,146,147]. Given its ability to cross the BBB, Lf presents promising therapeutic potential for neurodegenerative diseases.

### 7.2. Therapeutic Synergy

Lf’s potential extends beyond monotherapy, as combining it with other therapeutic agents can enhance its efficacy. Lf exhibits synergistic activity with various antiviral, antimicrobial, antifungal, and antitumor drugs [148,149,150,151,152,153]. The synergistic potential of Lf with other molecules has also been demonstrated in neurodegenerative diseases [154,155,156] (Table 2).

Lf-loaded curcumin nanoparticles potentially promoted neuroprotective activity in rotenone-treated SK-N-SH cells, demonstrating increased intracellular drug uptake, sustained retention, and antioxidant activity [154].

Lf-modified berberine nanoliposomes were used to treat a mouse model of AD, induced by the injection of amyloid-beta into the lateral ventricle. The study results demonstrated that the Lf modification enhanced the neuroprotective effects of berberine nanoliposomes, including inhibiting acetylcholinesterase, reducing tau over-phosphorylation, and preventing neuronal apoptosis. Additionally, the treatment resulted in a significant behavioral improvement in the mouse model [155].

Another study developed a quercetin delivery system encapsulated in liposomes modified with RMP-7 and Lf (RMP-7-Lf-QU-LS) to evaluate its ability to cross human brain microvascular endothelial cells (HBMECs) regulated by human astrocytes (HAs) and treat SK-N-MC cells after an insult with cytotoxic amyloid-beta fibrils. The results showed that incorporating RMP-7 and Lf into the particles increased the surface nitrogen level, slightly enhanced the system’s ability to cross the BBB, and reduced the transepithelial electrical resistance (TEER), indicating a slight opening of tight junctions. Additionally, RMP-7-Lf-QU-LS was able to reduce amyloid-beta-induced neurotoxicity and inhibit apoptosis [156].

Orexin is a neuropeptide that plays a crucial role in regulating the sleep-wake cycle, motivation, and feeding behavior [160]. In PD, increased levels of orexin-A have been correlated with anxiety, cognitive impairment, and other non-motor symptoms [160]. Although it has no specific application for PD, a study investigating the ability of orexin A to modulate mesolimbic dopamine transmission during feeding showed that lactoferrin-conjugated liposomes with orexin A enhanced dopamine release in the nucleus accumbens shell and facilitated orexin A delivery to the central nervous system, improving its bioavailability and therapeutic potential, as this peptide is large and does not cross alone the BBB [157].

The nitrogen-doped carbon dot (CD) is capable of neutralizing ROS and producing nitric oxide (NO) due to its nitrogen groups on the surface, which are attached to the sp2-hybridized π system [137]. Additionally, it can chelate iron ions, disrupting the catalytic iron cycle and inhibiting the Fenton reaction. The combination of Lf, polyethylene glycol (PEG), and CD (CD-PEG-Lf) enables non-destructive crossing of the BBB, targeting dopaminergic neurons via both NO-mediated reversible BBB opening and Lf receptor-mediated transport. This study demonstrated that CD-PEG-Lf acts as an antioxidant, reducing oxidative stress through iron chelation, free radical neutralization, and synergy with iron reflux prevention mediated by Lf. This approach significantly reduces brain inflammation and improves behavioral performance in PD mouse models [137].

A study developed PLGA nanoparticles loaded with resveratrol (RSV) and conjugated with Lf (Lf-RSV-PLGA-NPs) to enhance its brain bioavailability and neuroprotective effects in PD treatment [159]. The results showed that this formulation exhibited greater cellular uptake, reduced oxidative stress and MPP+-induced mitochondrial dysfunction, and increased RSV accumulation in the brain. In the experimental PD model, Lf-RSV-PLGA-NPs demonstrated superior neuroprotective effects compared to free RSV. These findings suggest that Lf-RSV-PLGA-NPs could be a promising strategy to enhance RSV efficacy in neurodegeneration treatment [159].

The combination of Lf with astaxanthin (ASX), a potent antioxidant and anti-inflammatory agent, in liposomes (Lf-ASX-LPs) significantly enhanced cellular uptake and permeability, increasing internalization in SH-SY5Y cells by 16.7 times compared to liposomes without Lf [138]. It was also observed that the presence of Lf in the liposomes improved brain penetration ability, increased membrane potential, and reduced cell loss, levels of ROS, and nitric oxide. Treatment with Lf-ASX-LPs in MPTP-induced PD mouse models resulted in increased dopamine levels, higher TH+ neuron density, improved behavior, and reduced neuroinflammation [138].

The PLGA@CAY@Lf nanoparticle, composed of poly (lactic-co-glycolic acid) (PLGA), Lf, and CAY10603 (CAY), a potent and selective histone deacetylase inhibitor, was developed to address the imbalance between histone acetylation and deacetylation, a disturbance characteristic of PD [139]. The results showed that treatment with PLGA@CAY@Lf normalized dopamine levels and tyrosine hydroxylase activity, reduced neuroinflammation, and improved behavioral deficits in mice with methamphetamine (Meth)-induced PD [139].

The Lf-modified organic-inorganic hybrid mesoporous silica nanoparticle system (Lf-lip@LC-MSNs) was designed to improve the brain delivery of levodopa and curcumin, enhancing their neuroprotective effects and facilitating more effective recovery of motor function in PD models [158]. Additionally, the system demonstrated low systemic toxicity, underscoring its potential as a promising therapeutic strategy for PD treatment. [158].

Lf demonstrates significant potential as monotherapy and when combined with other therapeutic agents, enhancing their efficacy across various disease models. By improving the delivery and effectiveness of neuroprotective compounds such as curcumin, berberine, and quercetin, Lf offers a promising avenue for advancing treatment strategies for conditions like AD and PD. These findings highlight Lf’s versatility and potential as a powerful tool in neuroprotection and beyond.

## 8. From Bench to Beside: Bridging Preclinical Promise with Clinical Reality

Lf has been studied in several clinical trials for its potential in various therapeutic areas, such as infectious diseases, gastrointestinal disorders, neurodegenerative diseases, and even as a supplement to improve overall health.

Lf is widely studied in infant formulas due to its potential to enhance gut and immune health in babies. Clinical trials have assessed its effects on newborns, preterm infants, and children, showing benefits in preventing infections and supporting immune development [161,162,163,164,165,166].

Other clinical applications of Lf are being investigated, highlighting its potential as a promising agent in treating iron deficiency anemia, especially in patients on regular hemodialysis [167]. Additionally, Lf has shown the ability to alleviate subjective symptoms, such as discomfort in the mouth and throat, commonly associated with low humidity [168]. Lf also provides significant benefits in supporting the immune system by maintaining plasmacytoid dendritic cell activity and preserving respiratory and systemic physical conditions in healthy adults, contributing to overall well-being improvement [169].

The role of Lf in AD was investigated in a pilot study involving 50 patients (28 men and 22 women) who were treated with 250 mg/day of Lf for three months. The results showed reduced serum levels of acetylcholine, serotonin, and antioxidant and anti-inflammatory markers, decreased Akt expression in peripheral blood lymphocytes (PBL), and lower levels of PI3K and p-Akt in PBL lysate. Similarly, elevated levels of Aβ 42, cholesterol, oxidative stress markers, IL-6, heat shock protein (HSP) 90, caspase-3, and p-tau, as well as increased expression of tau, MAPK1, and PTEN, were significantly reduced following Lf intake. The improvement in these AD surrogate markers observed after Lf treatment was reflected in enhanced cognitive function, as assessed by the Mini-Mental State Examination (MMSE) and Alzheimer’s Disease Assessment Scale-Cognitive Subscale 11-item (ADAS-COG 11) questionnaires, which were used as clinical endpoints [101].

These clinical studies are ongoing in various parts of the world, and Lf is being explored as a potential adjunct treatment or supplement for several conditions. However, further research is needed to confirm its effects and determine the ideal doses for different therapeutic indications.

## 9. Redefining Therapeutics: Beyond Parkinson’s Disease

### 9.1. Broader Applications in Synucleionophaties

The various functional applications of Lf place it in a position of great therapeutic potential for neurodegenerative diseases (Figure 4) [170]. The significant increase in studies on Lf and neurodegenerative diseases over the past decade reflects the growing interest in exploring its role in these conditions [7,171]. These studies have focused on investigating Lf’s involvement and its potential beneficial effects, expanding the understanding of the underlying mechanisms [88,130,131,132,133,134,135,136,137,138,139].

Lf has demonstrated significant potential in reducing oxidative damage and regulating the immune response, which could help slow the progression of neurodegenerative diseases, such as PD, by mitigating the toxic effects of aSyn buildup [37,83,87]. Its ability to interact with iron is particularly relevant as it could reduce iron-induced oxidative stress, a major contributor to neurodegeneration in synucleinopathies [7].

Lf plays an anti-inflammatory role in various conditions, including liver and respiratory diseases, obesity, and ocular disorders, and this property is also evident in neurodegenerative diseases [7]. Chronic neuroinflammation is a component of the neurodegenerative process in PD, characterized by microglial activation, which triggers the release of inflammatory cytokines such as IL-1, IL-2, TNF-α, IL-6, TGF-β, and IFN-γ [172]. The increased levels of these cytokines represent an immune system response to neural damage; however, they may also contribute to disease progression [173].

Numerous studies have highlighted the anti-inflammatory role of Lf in neurodegenerative diseases. Activated microglia in the central nervous system produce and release Lf during neuroinflammation. Lf has been shown to reduce the expression of pro-inflammatory cytokines such as IL-6, IL-1, and TNF-α [170]. Additionally, in AD, Lf helps inhibit inflammatory responses and mitigate oxidative damage [174]. This is achieved by suppressing inflammatory cytokines and modulating redox-active iron. Furthermore, Lf’s effects on ROS and inflammation may offer protection against AD by influencing key signaling pathways, such as the phosphorylated protein kinase B pathway and phosphatase and tensin homolog [7,174].

It is not yet fully understood whether the overexpression of Lf/LfRs in the brain is a contributing factor to the development of PD. One hypothesis suggests that the increase in lactoferrin levels and its receptors in PD, combined with a lack of an intracellular feedback loop, could promote iron accumulation [175,176]. However, further studies are needed to accurately determine the role of Lf in the neurodegeneration process, particularly with regard to iron accumulation.

However, several data suggest how Lf’s functional properties might alleviate the progression of PD (Figure 3 and Figure 4). First, Lf could interact with and bind excess extracellular Fe^2+^, albeit with lower affinity, and reduce its internalization via the DMT1 receptor and decrease the expression of DMT1 (Figure 3—step (2)). Additionally, Lf could be internalized through Lf receptors and bind to intracellular iron accumulation, minimizing its detrimental effects (Figure 3—step (4)). Known for its antioxidant activity, Lf has been reported in various studies to reduce ROS and oxidative stress, exerting a neuroprotective effect (Figure 3—steps (5), (7) and (10)). Moreover, Lf may also bind to monomeric aSyn, preventing its aggregation or even inducing the disaggregation of oligomeric species (Figure 3—step (9)). These properties suggest that Lf could offer a broader therapeutic benefit across a spectrum of neurodegenerative diseases involving aSy aggregation, offering hope for new treatment options in conditions with limited therapeutic strategies (Figure 4) [7].

### 9.2. Preventive Potential

Considering Lf’s natural presence in the body and its low toxicity profile, it emerges as a promising candidate for use as a prophylactic agent [86,88]. Individuals at high risk for synucleinopathies, whether due to genetic factors or identified through biomarker screening, may benefit from early intervention with Lf-based therapies. Such proactive approaches could delay or even prevent the onset of neurodegenerative diseases, providing a novel strategy for managing these conditions before significant symptoms develop. While no studies are currently underway specifically targeting this preventive use, the growing body of evidence showing Lf’s benefits in established diseases suggests it may also have a protective effect when used earlier in the disease course.

## 10. Challenges and Opportunities in the Lactoferrin Landscape: Overcoming Bioavailability Barriers

Despite its potential, Lf can face challenges related to stability and bioavailability, which hinder its therapeutic use [177,178]. Although it is a non-toxic protein capable of rapidly crossing the BBB, optimizing its bioavailability requires a better understanding of effective administration methods [2,179,180].

Oral administration is the safest and most convenient route for Lf, with approximately 60% of its content, especially holo-Lf, remaining structurally intact through the gastric stage [2]. Lf is absorbed across the apical membrane of intestinal cells by LfRs. However, its bioavailability varies significantly due to enzymatic degradation in the stomach and small intestine, as well as its short half-life in blood plasma [2]. Intranasal administration offers advantages such as rapid absorption, bypassing first-pass metabolism, and ease of application, but it is limited by the small amounts that can be delivered [2]. To address these challenges, nanocarrier-based delivery systems have been developed to protect Lf from enzymatic degradation, improving its stability, permeability, and overall bioavailability [181].

Strategies such as bioengineering Lf derivatives with enhanced stability or developing co-delivery systems may help overcome these limitations. Additionally, determining the ideal concentrations for treating neurodegenerative diseases is essential. To address these issues, more clinical studies are needed to gain deeper insights into how to maximize Lf’s therapeutic efficacy.

## 11. Conclusions: Lactoferrin’s Promise as a Disruptive Force in Neurodegeneration

Lf has been studied for many years, primarily due to its multifunctional nature, which makes it stand out for its various benefits. However, in the last decade, research on the relationship between Lf and neurodegenerative diseases has gained greater relevance, focusing on its potential neuroprotective effects.

Neurodegenerative diseases are primarily incurable, making Lf a promising candidate for treating complications, slowing the progression of these diseases, and preventing their onset. Several studies conducted in mouse models have shown interesting results, but its therapeutic potential in humans still lacks more robust clinical data.

Despite significant progress in Lf research and its potential applications in neurodegenerative diseases, several limitations remain before Lf can be considered an effective therapy. While studies in animal models show promising results, the lack of robust human clinical data is a major limitation. Additionally, the exact mechanism by which Lf exerts its neuroprotective effects, particularly in inhibiting aSyn aggregation, is not yet fully understood.

Therefore, more clinical and preclinical studies are needed to confirm Lf’s efficacy in treating neurodegenerative diseases, given the complexity of these conditions and the need for therapies addressing multiple aspects of neurodegeneration. Clinical trials are crucial for establishing its efficacy in human patients and evaluating its safety profile.

Challenges also exist in optimizing Lf formulation and delivery to the central nervous system. Optimizing Lf delivery to the CNS through innovative drug delivery systems, such as liposomes or nanoparticles, could improve its bioavailability and therapeutic outcomes.

Preventive applications of Lf in neurodegenerative diseases should also be explored, especially for individuals at high risk of developing conditions like Parkinson’s disease or Alzheimer’s disease. Investigating its role in the early stages of disease could open new avenues for delaying or even preventing disease onset. These directions will help clarify Lf’s potential as a viable therapeutic and preventive strategy for neurodegenerative disorders.

Although many challenges remain before Lf can be recognized as an effective therapy for diseases like Parkinson’s, the preliminary data available are encouraging, indicating a promising path for future investigations and clinical applications.

## Figures and Tables

**Figure 1 brainsci-15-00380-f001:**
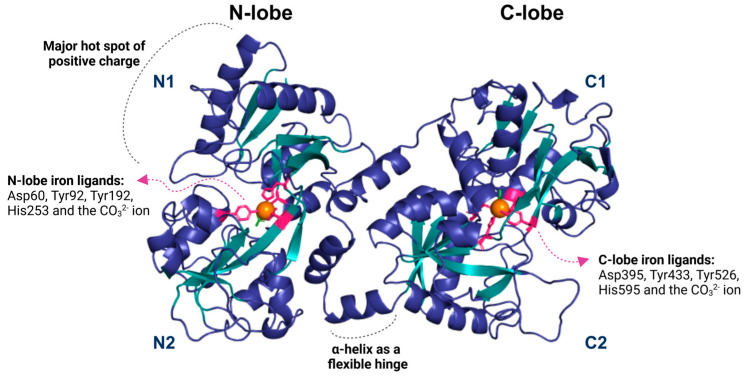
The lactoferrin structure is divided into two lobes, N and C, subdivided into two domains: N1 and N2 for the N lobe and C1 and C2 for the C lobe. Each lobe contains an iron-binding region (depicted as an orange sphere) that forms interactions with the residues aspartate, tyrosine, and histidine (shown in pink), as well as a carbonate ion (green). Lf contains a small helix that functions as a flexible hinge, allowing movement between its domains. The N1 domain of lactoferrin contains a predominantly positive region, possibly related to its interaction with various ligands. The image was generated using the Pymol 3.1 software with the PDB code 1BLF.

**Figure 2 brainsci-15-00380-f002:**
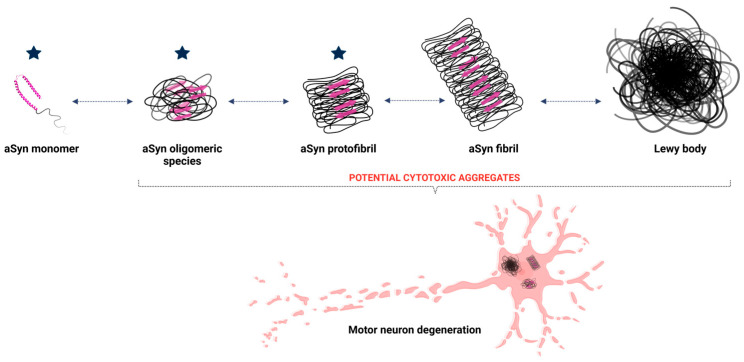
Aggregation of aSyn leading to Lewy body formation. The structure of aSyn consists of an alpha-helix and an intrinsically disordered region. During the aggregation process, monomeric aSyn undergoes structural changes, increasing the beta-sheet secondary structure (represented by pink arrows), facilitating the formation of oligomeric species, such as dimers, and progressing to fibrils. Once formed, these fibrils aggregate to form Lewy bodies, which are responsible for neuronal degeneration in Parkinson’s disease. The blue stars represent the potential stages at which lactoferrin may exert its function and minimize the impacts of the disease. Created with BioRender.com.

**Figure 3 brainsci-15-00380-f003:**
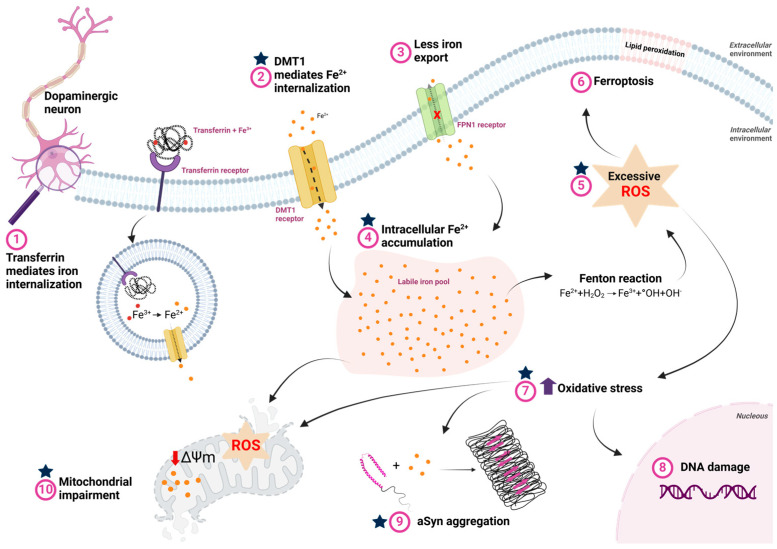
Iron metabolism dysfunction and its consequences in Parkinson’s disease. (1) Iron bound to transferrin is internalized via the transferrin receptor, where, in the process of endocytosis, Fe^3+^ is reduced to Fe^2+^ by endosomal reductases in the acidic environment of the endosome. Fe^2+^ is then transported to the cytosol by the divalent metal transporter (DMT1). (2) The extracellular excess of free iron, which occurs in PD, is also internalized through DMT1. (3) In PD, reduced ferroportin (FPN) activity impairs cellular iron efflux, contributing to iron accumulation. (4) The accumulated iron is stored in dynamic reserves known as the labile iron pool. (5) High intracellular iron concentrations facilitate interaction with hydrogen peroxide via Fenton reaction, increasing free radicals and oxidative stress. (6) Excess ROS promotes lipid peroxidation, which can lead to ferroptosis. (7) Furthermore, oxidative stress caused by elevated ROS due to intracellular iron overload also induces: (8) DNA damage, which can trigger neuronal cell death; (9) aSyn aggregation, favoring the formation of cytotoxic oligomeric species that contribute to cell death; and (10) mitochondrial dysfunction, characterized by increased ROS production, decreased membrane potential, and lipid peroxidation, further exacerbating neurodegeneration. The blue stars represent the potential stages at which lactoferrin may exert its function and minimize the impacts of the disease. Created with BioRender.com.

**Figure 4 brainsci-15-00380-f004:**
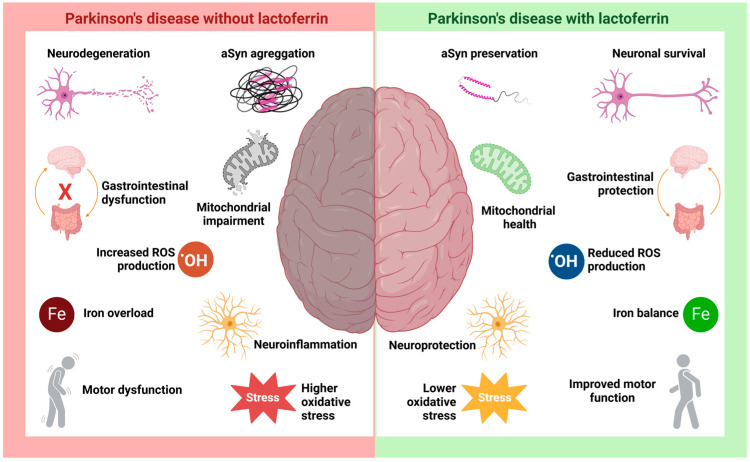
The hypothetical role of lactoferrin in Parkinson’s disease. PD is characterized by a range of well-studied pathological processes, such as neurodegeneration and neuroinflammation resulting from aSyn aggregation and Lewy body formation, along with intracellular iron accumulation, mitochondrial dysfunction, increased ROS production, and heightened oxidative stress. The disease is also associated with motor and gastrointestinal disturbances. Studies suggest that Lf may be crucial in modulating these pathological processes, promoting neuronal survival and neuroprotection. Its antioxidant activity is key in preserving mitochondrial function and mitigating damage caused by cellular stressors, such as iron accumulation and increased oxidative stress.

**Table 1 brainsci-15-00380-t001:** Summary of recent studies on the potential use of Lf in Parkinson’s disease across different models.

Study	Model	Lf Dose	Administration Route	Main Results
Rousseau et al., 2013 [84]	Midbrain cell cultures	Lf 0.1–30 µM	In Vitro treatment	Lf’s iron binding protects dopamine cells under oxidative stress; Lf accumulation in PD brains may reflect the brain’s protective response.
Wang et al., 2015 [83]	Ventral mesencephalon neurons (incubated with MPP+)	Apo- and holo-Lf 100 ng/mL	In Vitro treatment	Apo-Lf and holo-Lf are neuroprotective against MPP+; increase mitochondrial membrane potential; improve Cu/Zn-superoxide dismutase activity; enhance BCL-2 expression.
Xu et al., 2019 [85]	Mouse model of PD (MTPT injection)	Human Lf 4 mg/kg body weight	Intraperitoneal injection	Lf promove reduction of MPTP-induced apoptosis of dopaminergic neurons; decrease in neuroinflammation and histological alterations; suppression of excessive iron accumulation; downregulation of DMT1 and TFR; improvement in antioxidant enzyme activity.
Liu et al., 2020 [37]	Mouse model of PD (MTPT injection)	Apo- and holo-Lf 5, 10 and 15 mg/kg	Intragastric gavage	Lf treatment downregulated DMT1 and upregulated ferroportin 1; alleviated MPTP-induced accumulation of nigral iron; reduced serum iron and ferritin levels; decreased spleen iron content and spleen weight loss.
Kopaeva et al., 2021 [86]	Mouse model of PD (MTPT injection)	Human Lf 4 mg/animal	Intraperitoneal injection	Lf reduced MPTP toxicity; improvement in motor function and exploratory behavior; partial recovery of dopaminergic neurons in the substantia nigra; increase in TH-positive fibers in the striatum; evidence of neuroprotective and compensatory mechanisms.
Xu et al., 2024 [87]	Mouse model of PD (MTPT injection) and knockout of the astrocyte Lf gene	-	-	MPTP-treated astrocytic Lf knockout mice exhibited abnormal levels of effects implicated in glutamate and calcium homeostasis; mitochondrial dysfunction; and signs of oxidative stress.
Yong et al., 2024 [88]	Cellular model of PD (differentiating SH-SY5Y to dopaminergic neurons and exposure to rotenone)	Lf 1–10 µg/mL	In Vitro treatment	Lactoferrin pre-treatment reduced cell viability loss; prevented mitochondrial membrane potential impairment; decreased ROS generation; reduced pro-apoptotic activities (caspase activation and nuclear condensation); decreased Bax:Bcl2 ratio; increased pAkt expression.

**Table 2 brainsci-15-00380-t002:** Summary of recent studies showing the potential of Lf as a key player in enhancing the efficacy of various therapeutic molecules in treating neurodegenerative diseases.

Study	Model	Related-Disease	Lf Interaction	Administration Route	Main Results
Bollimpelli et al.,2016 [154]	Cell line SK-N-SH	PD	Curcumin loaded Lf nano particles prepared by sol-oil chemistry.	In Vitro treatment	Higher intracellular drug uptake; sustained drug retention; greater neuroprotective activity; reduced ROS levels.
Kuo & Tsao, 2017 [156]	Cell line SK-N-MC	AD	Quercetin, encapsulated liposomes grafted with RMP-7 and LF.	In Vitro treatment	Inhibited cell apoptosis and the expression of phosphorylated proteins associated with apoptosis; low toxicity; increased viability of SK-N-MC cells and reduced neurotoxicity induced by β-amyloid fibrils.
Lai et al., 2018 [157]	Male Sprague– Dawley rats	Not related disease	Orexin A-loaded lactoferrin-conjugated Liposomes.	Intravenous injection	Lactoferrin-conjugated liposomes with orexin A enhanced dopamine release in the nucleus accumbens shell; facilitate orexin A delivery to the central nervous system.
Wang et al., 2023 [155]	Mouse model of AD (Aβ-injected)	AD	Lf-modified berberine nanoliposomes.	Injection via caudal vein	Improved mouse behavior; reduced tau over-phosphorylation; inhibited acetylcholinesterase activity; and enhanced neuroprotective effects.
Guo et al., 2024 [158]	Mouse model of PD (MTPT injection)	PD	Carbon dots, polyethylene glycol and Lf.	Intravenous injection	Antioxidant; reduction of oxidative stress; reduction of brain inflammation; behavioral improvement.
Katila et al., 2022 [159]	Mouse model of PD (MTPT injection); SH-SY5Y and HBMECs cells	PD	Lf-conjugated resveratrol-loaded PLGA nanoparticles conjugated with Lf.	Injection via caudal vein	Increased internalization in SH-SY5Y and brain endothelial cells; Reduction of oxidative stress; improved brain bioavailability; enhanced protective effects in the MPTP-induced PD model.
Pham et al., 2025 [139]	HBMECs and SH-SY5Y cells; C57BL/6J mice (methamphetamine PD model) and female BALB/c	PD	Lf-decorated CAY10603-loaded poly(lactic-co-glycolic acid) nanoparticles.	Injection via caudal vein	Enhanced BBB penetration; restoration of acetylation balance; reversed mitochondrial dysfunction; suppressed ROS; inhibited aSyn accumulation; normalized dopamine and tyrosine hydroxylase levels; improved behavioral impairments in the Meth-induced PD mouse model.
Nguyen et al., 2025 [138]	Cell line SH-SY5Y and C57BL/6 mice (MTPT injection)	PD	Lf-conjugated astaxanthin-loaded liposomes.	Intravenous injection	Cytoprotective effects in vitro; liposomes demonstrated significantly improved cellular uptake; neuroprotective effects in the MPTP mouse model; alleviated behavioral impairments.
Guo et al., 2025 [158]	Mouse model of PD (MTPT injection) and SH-SY5Y cells	PD	LF-modified silica nanoparticles for co-delivery of levodopa and curcumin.	Intraperitoneal injection	Reduction of oxidative stress; lower aSyn accumulation; increased neuronal survival; optimized brain delivery; improvement of motor function; low systemic toxicity.

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
