# Peer review of "Lactoferrin as a Candidate Multifunctional Therapeutic in Synucleinopathies"

_brainsci, 2025, doi:10.3390/brainsci15040380_

Round 1
Reviewer 1 Report
Comments and Suggestions for Authors
The manuscript (brainsci-3557996) reviewed recent progress on lactoferrin (Lf) mainly in PD but also AD including its interaction with αSyn aggregation, iron chelation, modulation of oxidative stress and neuroinflammation, and its potential impact on the gut-brain axis. It was concluded that Lf could be a compelling candidate for future research and clinical development in neurodegenerative diseases.
Overall, the topic is interesting and the manuscript adds to the increasing literature on the discussion of potential neuroprotective benefit of Lf in neurodegenerative conditions including PD. I have the following comments for the authors:
- The layout of the sections and subsections is confusing, with many sections have only one subsection, e.g., 1.1, 4.1, etc.; the title of section 1.1 is the same as that for section 2; section 4 says “Lactoferrin and Neurodegenerative Disorders” but only PD is covered whereas section 5 is on “Lf in Other Neurodegenerative Disorders”.
- The review focuses on Lf in PD. Perhaps it would be helpful to be more critical in reviewing the literature on Lf in PD models. A table can be provided in section 4.1 listing the studies, outlining the information on the animal model, Lf dose, route of administration, detection methods, major findings, and authors’ comments/interpretation.
- Reference for Serrano et al. (2020) on page 2 line 84 is missing.
- Reference for Zhou et al., 2021 on page 11 line 434 is missing.
- What are possible adverse effects of Lf in PD therapy, eg, see Qian et al. (Lactoferrin/lactoferrin receptor: Neurodegenerative or neuroprotective in Parkinson's disease? Ageing Res Rev. 2024 Nov;101:102474)?
- Is levels of Lf increased in PD, e.g, see Hamm-Alvarez et al. (Levels of oligomeric α-Synuclein in reflex tears distinguish Parkinson's disease patients from healthy controls. Biomark Med. 2019 Dec;13(17):1447-1457)?
- The following publications can be added to Table 1: Guo et al. 2025 Lactoferrin-modified organic-inorganic hybrid mesoporous silica for co-delivery of levodopa and curcumin in the synergistic treatment of Parkinson's disease. Phytomedicine. 2025 Feb 20;140:156547; Katila et al Enhancement of blood-brain barrier penetration and the neuroprotective effect of resveratrol. J Control Release. 2022 Jun;346:1-19.
Author Response
We thank the reviewer for your thoughtful and constructive comments, which helped improve the quality and clarity of our manuscript. Below is our point-by-point response to the reviewer’s suggestions. All changes have been made in the revised manuscript. We have highlighted all the changes in yellow in the new document for easy identification.
- The layout of the sections and subsections is confusing, with many sections have only one subsection, e.g., 1.1, 4.1, etc.; the title of section 1.1 is the same as that for section 2; section 4 says “Lactoferrin and Neurodegenerative Disorders” but only PD is covered whereas section 5 is on “Lf in Other Neurodegenerative Disorders”.
Response: We have revised the section and subsection layout for improved clarity. Headings were updated to ensure no redundant or standalone subsections.
- The review focuses on Lf in PD. Perhaps it would be helpful to be more critical in reviewing the literature on Lf in PD models. A table can be provided in section 4.1 listing the studies, outlining the information on the animal model, Lf dose, route of administration, detection methods, major findings, and authors’ comments/interpretation.
Response: A new table summarizing key studies on Lf in PD models has been added to Section 4. The table includes the animal model, Lf dose, route of administration, detection methods and main findings.
- Reference for Serrano et al. (2020) on page 2 line 84 is missing.
Response: The missing reference for Serrano et al. (2020) has been added.
- 4. Reference for Zhou et al., 2021 on page 11 line 434 is missing
Response: The reference for Zhou et al. (2021) has been inserted appropriately in the text.
- 5. What are possible adverse effects of Lf in PD therapy, eg, see Qian et al. (Lactoferrin/lactoferrin receptor: Neurodegenerative or neuroprotective in Parkinson's disease? Ageing Res Rev. 2024 Nov;101:102474)?
Response: We now include a discussion of the potential adverse effects of Lf in Section 4, referencing Qian et al. (2024) and evaluating the neurodegenerative vs. neuroprotective roles of Lf in PD. Page 16.
- Mention Lf levels in PD patients (e.g., Hamm-Alvarez et al., 2019).
Response: We have added a reference to Hamm-Alvarez et al. (2019) and discussed the increase in Lf levels in reflex tears of PD patients, supporting its potential role as a biomarker. Page 8.
- The following publications can be added to Table 1: Guo et al. 2025 Lactoferrin-modified organic-inorganic hybrid mesoporous silica for co-delivery of levodopa and curcumin in the synergistic treatment of Parkinson's disease. Phytomedicine. 2025 Feb 20;140:156547; Katila et al Enhancement of blood-brain barrier penetration and the neuroprotective effect of resveratrol. J Control Release. 2022 Jun;346:1-19.
Response: These references have been included in Table 1, and the text has been updated accordingly.
Reviewer 2 Report
Comments and Suggestions for Authors
Barros et al elaboate on the significance of lactoferrin in synucleinopathies. I have the following comments regarding the work:
- Authors could elaborate more extensively on the pathophysiology of synucleinopathies. Certain factors impacting its course could be more stressed. - Ref. (A) Systemic Administration of Orexin a Loaded Liposomes Potentiates Nucleus Accumbens Shell Dopamine Release by Sucrose Feeding. Front Psychiatry. 2018;9:640. Published 2018 Dec 3. doi:10.3389/fpsyt.2018.00640 (B) Role of orexin in pathogenesis of neurodegenerative parkinsonisms. Neurol Neurochir Pol. 2023;57(4):335-343. doi:10.5603/PJNNS.a2023.0044
- The issue of alpha-synuclein aggregation could be more extensively presented in the context of particular entities in this group.
- The review could be presented in a more critical manner, additionally highlighting the limitations.
- The significance of lactoferrin in the context of inflammation could be more emphasized in the context of theories regarding the pathogenesis of neurodegenerative diseases.
- An overview on the use of lactoferrin in future perspectives would be valuable
Author Response
We thank the reviewer for your thoughtful and constructive comments, which helped improve the quality and clarity of our manuscript. Below is our point-by-point response to the reviewer’s suggestions. All changes have been made in the revised manuscript. We have highlighted all the changes in yellow in the new document for easy identification.
- Authors could elaborate more extensively on the pathophysiology of synucleinopathies. Certain factors impacting its course could be more stressed. - Ref. (A) Systemic Administration of Orexin a Loaded Liposomes Potentiates Nucleus Accumbens Shell Dopamine Release by Sucrose Feeding. Front Psychiatry. 2018;9:640. Published 2018 Dec 3. doi:10.3389/fpsyt.2018.00640 (B) Role of orexin in pathogenesis of neurodegenerative parkinsonisms. Neurol Neurochir Pol. 2023;57(4):335-343. doi:10.5603/PJNNS.a2023.0044
Response: We expanded Section 7.2 to include including studies on orexin and lactoferrin. Page 13.
- The issue of alpha-synuclein aggregation could be more extensively presented in the context of particular entities in this group.
Response: We elaborated on the clinical differences between PD, DLB, and MSA in relation to alpha-synuclein aggregation in Section 3.
- The review could be presented in a more critical manner, additionally highlighting the limitations.
Response: A new paragraph was added at the Conclusion to address limitations in current studies and the gaps in knowledge regarding Lf’s mechanism of action.
- The significance of lactoferrin in the context of inflammation could be more emphasized in the context of theories regarding the pathogenesis of neurodegenerative diseases.
Response: We expanded the discussion on Lf’s anti-inflammatory role, especially in the context of neurodegeneration, and its impact on glial activation and cytokine regulation. Page15/16.
- 5. An overview on the use of lactoferrin in future perspectives would be valuable
Response: A new text has been added outlining future directions, including potential clinical trials, delivery optimization, and preventive applications in Conclusion.
Reviewer 3 Report
Comments and Suggestions for Authors
This review introduces lactoferrin (Lf) as a neuroprotective agent and its relevance to neurodegenerative diseases, particularly Parkinson’s disease (PD).
This paper is well-structured, informative, and written in a fluent manner. Below are some minor suggestions that may help improve its quality:
- Could the authors provide a more detailed explanation of how lactoferrin (Lf) crosses the blood-brain barrier (BBB)? More attention should be given to the research on Lf receptors (LfR).
- As a protein, how does Lf avoid degradation and get absorbed through the intestine directly into the bloodstream? This aspect requires further elaboration.
- The discussion on co-delivery systems is rather brief. Given that many studies have explored this area, and considering its significance in the clinical application of Lf, more discussion is warranted.
- The research findings in Sections 5.1 and 5.2 could be better summarized in a table for clarity.
- In Table 1, the mouse model should explicitly state the specific modeling methods. Additionally, the administration route should be clearly indicated (e.g., whether it is via gavage).
Author Response
We thank the reviewer for your thoughtful and constructive comments, which helped improve the quality and clarity of our manuscript. Below is our point-by-point response to the reviewer’s suggestions. All changes have been made in the revised manuscript. We have highlighted all the changes in yellow in the new document for easy identification.
- Could the authors provide a more detailed explanation of how lactoferrin (Lf) crosses the blood-brain barrier (BBB)? More attention should be given to the research on Lf receptors (LfR).
Response: We expanded the section on BBB crossing (Section 7.1) to include more detail on Lf receptors and their role in receptor-mediated transcytosis.
- As a protein, how does Lf avoid degradation and get absorbed through the intestine directly into the bloodstream? This aspect requires further elaboration.
Response: We added a paragraph explaining how Lf resists proteolytic degradation and is absorbed in the intestine, citing relevant transport mechanisms and administration. Page 17.
- The discussion on co-delivery systems is rather brief. Given that many studies have explored this area, and considering its significance in the clinical application of Lf, more discussion is warranted.
Response: The discussion on co-delivery systems was expanded in Section 7, including more examples from the recent literature.
- 4. The research findings in Sections 5.1 and 5.2 could be better summarized in a table for clarity.
Response: We appreciate the reviewer’s suggestion regarding including a table summarizing section 5. However, we respectfully chose not to include such a table, as our review's primary focus is Synucleinopathies. While we briefly mention related neurodegenerative disorders for context, the central aim of the manuscript is to provide a discussion of mechanisms and therapeutic perspectives related specially to Parkinson’s disease.
- 5. In Table 1, the mouse model should explicitly state the specific modeling methods. Additionally, the administration route should be clearly indicated (e.g., whether it is via gavage).
Response: Table 1 was revised to specify the modeling methods (e.g., MPTP, rotenone) and administration routes (e.g., oral, intranasal, gavage) used in each study.
Round 2
Reviewer 2 Report
Comments and Suggestions for Authors
Authors have sufficiently improved the manuscript.
Reviewer 3 Report
Comments and Suggestions for Authors
It is good enough for publication.